# Dickkopf-Related Protein 1 as Response Marker for Transarterial Chemoembolization of Hepatocellular Carcinomas

**DOI:** 10.3390/cancers14194807

**Published:** 2022-09-30

**Authors:** Anne Olbrich, Olga Gros, Sebastian Ebel, Timm Denecke, Holger Gößmann, Nicolas Linder, Florian Lordick, Dirk Forstmeyer, Daniel Seehofer, Robert Sucher, Sebastian Rademacher, Johannes Niemeyer, Madlen Matz-Soja, Thomas Berg, Florian van Bömmel

**Affiliations:** 1Division of Hepatology, Department of Medicine II, Leipzig University Medical Center, 04103 Leipzig, Germany; 2Helios Clinic Köthen, Department of Anesthesia and Intensive Care, 06366 Köthen, Germany; 3Department of Diagnostic and Interventional Radiology, Leipzig University Medical Center, 04103 Leipzig, Germany; 4University Liver Tumor Center (ULTC), Leipzig University Medical Center, 04103 Leipzig, Germany; 5University Cancer Center Leipzig (UCCL), Leipzig University Medical Center, 04103 Leipzig, Germany; 6Department of Visceral, Vascular, Thoracic and Transplant Surgery, Leipzig University Medical Center, 04103 Leipzig, Germany; 7Rudolf-Schönheimer-Institute for Biochemistry, University of Leipzig, 04103 Leipzig, Germany

**Keywords:** HCC, DKK-1, AFP, personalized treatment, treatment response

## Abstract

**Simple Summary:**

The response of hepatocellular carcinomas (HCC) to transarterial chemoembolization (TACE) is variable. In view of the dynamic development of treatment options for HCCs, an early selection of the most effective therapy for maximizing individual treatment success has a high medical need. We show that serum levels of circulating Dickkopf-related protein 1 (DKK-1) are associated with a 12-week response to subsequent TACE in European patients. DKK-1 levels also allowed for identification of responders in patients with normal levels of alpha fetoprotein. Our findings are a step toward developing DKK-1 as a novel HCC response marker and an impulse to investigate the mechanisms underlying the treatment response of HCCs.

**Abstract:**

Background and Aims: In the treatment of hepatocellular carcinoma (HCC), response prediction to transarterial chemoembolization (TACE) based on serum biomarkers is not established. We have studied the association of circulating Dickkopf-related protein 1 (DKK-1) with baseline characteristics and response to TACE in European HCC patients. Methods: Patients with HCC treated with TACE from 2010 to 2018 at a tertiary referral hospital were retrospectively enrolled. Levels of DKK-1 were measured in serum samples collected before TACE. Response was assessed according to mRECIST criteria at week 12 after TACE. Results: Ninety-seven patients were enrolled, including seventy-nine responders and eighteen refractory. Before TACE, median DKK-1 serum levels were 922 [range, 199–4514] pg/mL. DKK-1 levels were lower in patients with liver cirrhosis (*p* = 0.002) and showed a strong correlation with total radiologic tumor size (r = 0.593; *p* < 0.001) and with Barcelona Clinic Liver Cancer stages (*p* = 0.032). Median DKK-1 levels were significantly higher in refractory patients as compared to responders (1471 pg/mL [range, 546–2492 pg/mL] versus 837 pg/mL [range, 199–4515 pg/mL]; *p* < 0.001), and DKK-1 could better identify responders than AFP (AUC = 0.798 vs. AUC = 0.679; *p* < 0.001). A DKK-1 cutoff of ≤1150 pg/mL was defined to identify responders to TACE with a sensitivity of 78% and specificity of 77%. DKK-1 levels were suitable to determine response to TACE in patients with low AFP serum levels (AFP levels < 20 ng/mL; AUC = 0.843; 95% CI [0.721–0.965]; *p* = 0.003). Conclusion: DKK-1 levels in serum are strongly associated tumor size and with response to TACE in European HCC patients, including those patients with low AFP levels.

## 1. Introduction

Hepatocellular carcinoma (HCC) is one of the most frequent malignancies and a rising cause of cancer-related deaths worldwide [1]. According to the widely used Barcelona Clinic Liver Cancer (BCLC) staging system for patients with intermediate stages (BCLC B) or with multiple lesions, transarterial chemoembolization (TACE) is a standard first-line treatment [2,3,4]. Hereby, TACE shows variable six month response rates between 20 and 45% [5,6,7]. However, TACE may also be used in patients with early (BCLC A), e.g., patients for whom curative treatment is not feasible owing to various clinical factors (i.e., those with a solitary nodule or up to three nodules under 3 cm) or for patients waiting for liver transplantation, and also in those patients with advanced HCC (BCLC C) that are mostly allocated to systemic treatment [5,6,7,8]. Therefore, to facilitate individual optimized treatment strategies, the development of biomarkers for patients with different equivalent treatment options has a high medical need.

Currently, alpha-fetoprotein (AFP) in serum is used as response marker to HCC treatment in some situations, but it has not been validated as a response prediction marker [9,10]. Apart from AFP, no circulating biomarkers for HCC treatment monitoring have been established yet. Recent studies have shown that serum levels of Dickkopf-related protein 1 (DKK-1), a circulating intermediate of the Wnt/β-catenin signaling cascade that is overexpressed in HCC, are associated with poor clinical outcome, making it an interesting biomarker candidate for HCC treatment monitoring [11,12,13,14,15]. Interestingly, DKK-1 levels before and during TACE were previously found to be associated with response to TACE in Asian patients who, however, were predominantly hepatitis B surface antigen positive [16,17]. It is unclear whether DKK-1 levels show a similar association in European patients that have a different genetic and different tumor etiology. We aimed to investigate the association of serum DKK-1 levels with tumor and patient characteristics in a European population and to characterize its role as a biomarker for TACE treatment outcomes.

## 2. Patients and Methods

### 2.1. Patients

For our retrospective cohort study, all patients over 18 years with HCC treated with TACE at Leipzig University Medical Center between 2010 and 2018 were assessed for inclusion. Unequivocal diagnosis of HCC by radiologic criteria, imaging-based tumor response assessment at 12 weeks after treatment initiation, availability of a serum sample stored at –20 °C and collected at the start of the therapy, and written informed consent of the patients were mandatory for study inclusion. Patients were excluded if they had another tumor entity in addition to HCC. 

### 2.2. HCC Diagnosis and Treatment Evaluation

The diagnosis of HCC and the treatment response were assessed based on contrast-enhanced multiphase computed tomography (CT) and/or magnetic resonance imaging (MRI) according to current treatment guidelines [2]. Patients were diagnosed with HCC if their tumor had typical features of HCC (i.e., hypervascularity in the arterial phase and washout in the portal venous or delayed phase). Tumor stages were defined according to the BCLC staging system. Treatment allocation was based on a multidisciplinary tumor board decision.

Treatment efficacy was evaluated 12 weeks after TACE based on mRECIST criteria. Responders were defined as patients with complete, partial response, or stable disease of the lesions treated with TACE, while refractoriness in the treated liver region was defined as either progressive disease, viable lesion > 50%, tumor revascularization of the treated lesions, or appearance of new hypervascularized lesions [18]. 

### 2.3. Quantification of DKK-1 and AFP

DKK-1 and AFP were quantified from serum samples collected at the time point of treatment initiation and stored −20 °C. DKK-1 was measured by quantitative solid-phase ELISA (Quantikine ELISA Human DKK-1 Immunoassay, DKK-100, R&D Systems, Minneapolis, USA, lower detection limit 15.6 pg/mL) according to manufacturer’s instructions using a fully automated microtiter plate analyzer ETI-Max 3000 (DiaSorin, Saluggia, Italy). Final DKK-1 serum concentrations were obtained by interpolation from a standard curve and expressed in pg/mL. Two analyses per sample were carried out, and the average was calculated. AFP was measured in the serum by Fujifilm Wako Chemicals Europe (Neuss, Germany, lower detection limit 0.03 ng/mL).

### 2.4. Statistics

The patient population was divided into groups by different definitions of low AFP: (1) AFP <130 ng/mL [19], (2) AFP < 20 ng/mL [20], and (3) AFP < 8 ng/mL (internal standard). 

IBM SPSS Statistics software version 25 was used for data analyses. A *p*-value less than 0.05 was considered significant. For the description of continuous variables, mean and standard deviation or median and interquartile range were used as appropriate, while for the description of qualitative variables, absolute frequencies and percentages were used. Differences between two independent groups were tested with the Mann–Whitney U test. Pearson correlation coefficients were used to calculate the correlation between variables. Receiver operating characteristics (ROC) curves were constructed to assess sensitivity, specificity, and respective areas under the curves (AUCs) with 95% CI. To determine the best cutoff point for therapy response, the highest Youden’s index was calculated. Survival curves were constructed by Kaplan–Meier method. Univariate and multivariate logistic regression analyses were used to screen for independent predictors for response. For validation of the predictive response value of DKK-1, the cohort was randomly divided into a training cohort and a validation cohort by random sampling at a ratio of 2:1 and analyzed.

## 3. Results

### 3.1. Patient Selection and Baseline Characteristics

From a total of 977 patients diagnosed with HCC between 2010 and 2018, 546 had received TACE as a first line treatment and contrast-enhanced multiphase CT or MRI performed at week 12 after treatment was available. A total of 449 patients were excluded due to a lack of a baseline serum sample, existence of other tumor entities, or absence of written consent. The resulting population consisted of 97 patients (median age 63.0 [range, 31–83 years], 86 males) (Figure 1). Baseline characteristics of the study population are shown in Table 1. 

### 3.2. Treatment Response and Overall Survival

In the overall cohort, the radiologic response rate to TACE after 12 weeks was 81% (79/97 patients) (Figure 1). The median total follow-up time was 15 months [range, 0–89 months], and 33 patients died during the observation period (BCLC-stages A/B/C/D: n = 13/13/6/1, respectively). The immediate cause of death was tumor progression in (n = 23), cirrhosis complications (n = 7), treatment related complications (n = 1), and infectious complications (n = 2). During the observation period, 82 patients received additional transarterial treatment, 27 patients underwent orthotopic liver transplantation (OLT), and 17 patients received systemic treatments.

### 3.3. DKK-1 and AFP Levels before Treatment

In the study population, median DKK-1 levels were 922 [range, 199–4514] pg/mL, and median AFP levels were 13 [range, 1–13019] ng/mL. DKK-1 and AFP levels before treatment showed a weak correlation (r = 0.022; p = 0.832) (Appendix A).

### 3.4. Association of DKK-1 Levels with Patient Characteristics

DKK-1 levels were significantly higher in patients without cirrhosis (n = 9) as compared to patients with cirrhosis (n = 88) (1705 [range, 825–4515] pg/mL versus 888 [range, 199–2313] pg/mL; *p* = 0.002) (Figure 2A). Interestingly, the DKK-1 level at baseline strongly correlated with the number of platelets (r = 0.802; *p* < 0.001), and moderately with neutrophiles (r = 0.598; *p* < 0.001) and leukocytes (r = 0.581; *p* < 0.001). In addition, a weak negative correlation was found between DKK-1 and bilirubin (r = −0.329; *p* < 0.001) and albumin (r = 0.294; *p* = 0.003) (Appendix A). No correlations were found between the serum DKK-1 level at baseline and CRP level or alanine aminotransferase. Male (n = 86) and female patients (n = 11) showed similar median DKK-1 levels (933 [range, 199–4515] pg/mL versus 825 [range, 403–1888] pg/mL; *p* = 0.794). There was no association between DKK-1 levels and the age of the patients (r = 0.032; *p* = 0.754).

### 3.5. Association of DKK-1 Levels with Tumor Distribution

The correlation between tumor size analyzed by CT or MRI imaging and DKK-1 before TACE showed a strong correlation (r = 0.593; *p* < 0.001) (Figure 2B). Moreover, patients with HCC lesions >5 cm (n = 29) showed significantly higher DKK-1 serum levels compared to patients with lesions <5 cm (n = 68) (1261 [range, 403–4515] pg/mL versus 838 [range, 199–2263] pg/mL; *p* < 0.001). DKK-1 levels were similar between patients with bilobar tumor localization (n = 24) compared to single lobe localization (n = 73) (918 [range, 367–4515] pg/mL versus 930 [range, 199–2263] pg/mL; *p* = 0.196). In addition, DKK-1 levels were significantly higher in patients in BCLC stage C/D (n = 10) as compared to stage A (n = 54) (1540 [range, 515–4515] pg/mL versus 848 [range, 199–2013] pg/mL; *p* = 0.032) (Figure 2C). 

### 3.6. Association of DKK-1 and AFP Levels with Response to TACE

Median DKK-1 levels were significantly higher in patients that were refractory to TACE as determined at week 12 after treatment (n = 18) as compared to responders (n = 79) (1471 [range, 546–2492] pg/mL versus 837 [range, 199–4515] pg/mL; *p* < 0.001) (Figure 3A). Similarly, median AFP at baseline was higher in refractory patients (49 [range, 4–5895] ng/mL versus 10 [range, 1–13019] ng/mL; *p* = 0.018) (Figure 3B). DKK-1 levels had a stronger association with response to TACE (AUC = 0.798; 95% CI [0.692–0.905]; *p* < 0.001) as compared to AFP (AUC = 0.679; 95%C I [0.549–0.809]; *p* = 0.018) (Figure 3C). The optimum cutoff level for DKK-1 for discriminating responders to TACE from refractory patients was 1150 pg/mL with a sensitivity of 78% and a specificity of 77%.

The association of DKK-1 levels with response to TACE could be confirmed across patients in different BCLC stages. Thus, in patients in BCLC stage A (n = 6) that were refractory to TACE, median DKK-1 levels were significantly higher as compared to responders (n = 48) (1391 [range, 847–1965] pg/mL versus 818 [range, 199–2013] pg/mL; *p* = 0.01). Median DKK-1 levels were also significantly higher in refractory patients with BCLC stage B (n = 10) as compared to responders with BCLC stage B (n = 26) (1341 [range, 546–2313] pg/mL versus 830 [range, 305–2284] pg/mL; *p* = 0.023) (Figure 3D). Using univariate and multivariate regression (likelihood ratio: forward stepwise), we were able to show that DKK-1 at baseline is an independent predictor for response after TACE (Appendix A). 

For validation of the predictive response value of DKK-1, our cohort was randomly divided into a training cohort (n = 65) and an internal validation cohort (n = 32) by random sampling at a ratio of 2:1. We were able to show that the DKK-1 value before TACE was significant in both random cohorts (*p* = 0.002 in the training cohort and *p* = 0.012 in the validation cohort, Appendix A). In addition, ROC analyses showed that DKK-1 outperformed AFP in response prediction. DKK-1 levels had a stronger association with response to TACE (AUC = 0.794; 95% CI [0.664–0.924]; *p* = 0.002) as compared to AFP (AUC = 0.642; 95% CI [0.476–0.809]; *p* = 0.126) in the training cohort as well as in the validation cohort (AUC = 0.833; 95% CI [0.667–1]; *p* = 0.012) as compared to AFP (AUC = 0.782; 95% CI [0.622–0.942]; *P* = 0.034; Appendix A).

### 3.7. DKK-1 and Treatment Response in Patients with Low AFP Levels

The association of DKK-1 with response to TACE was analyzed in 80 patients (82%) with low AFP levels defined by AFP cutoffs of 130 ng/mL, 20 ng/mL, and 8 ng/mL. In patients with AFP levels < 130 ng/mL, median DKK-1 levels in responders (n = 68) and refractory patients (n = 12) were 868 [range, 199–2284] pg/mL versus 1391 [range, 546–2313] pg/mL (*p* = 0.005) (Figure 4A). ROC analyses showed an AUC of 0.756 [0.613–0.900] (Figure 4D), and the optimum response cutoff for DKK-1 for discriminating responders was 1150 pg/mL (sensitivity 75%, specificity 75%). In patients with AFP levels <20 ng/mL, median DKK-1 levels in responders (n = 51) and refractory patients (n = 7) were 840 [range, 199–2284] pg/mL versus 1646 [range, 936–2313] pg/mL (*p* = 0.003). The optimum response cutoff for DKK-1 for discriminating responders was also 1150 pg/mL (sensitivity 86%, specificity 77%, AUC = 0.756 [0.613–0.900]) (Figure 4B,E). In patients with AFP levels <8 ng/mL, median DKK-1 levels in responders (n = 37) and refractory patients (n = 5) were 840 [range, 305–2284] pg/mL versus 1646 [range, 936–2313] pg/mL (*p* = 0.023) (Figure 4C). For discriminating responders, the optimum response cutoff for DKK-1 was 933 pg/mL (sensitivity 100%, specificity 60%; AUC = 0.843 [0.721–0.965]; Figure 4F and Table 2).

### 3.8. Association of DKK-1 Levels with Survival 

Patients with DKK-1 serum levels above the calculated cutoff for response of 1150 pg/mL showed a shorter overall survival than patients with DKK-1 serum levels below 1150 pg/mL (Figure 5). However, the differences were not significant (*p* = 0.084).

## 4. Discussion

In the present study, we have assessed the association of serum DKK-1 levels with patient and tumor characteristics as well as with response to TACE and survival after TACE in a European population with HCC for the first time. We found that DKK-1 serum levels were significantly lower in patients with liver cirrhosis, and that they correlated with serum markers of liver function, such as albumin, bilirubin, and platelets. DKK-1 serum levels correlated strongly with radiological total tumor diameter and with BCLC stages. Serum levels of DKK-1 had a strong association with 12-week response to TACE. We could define a DKK-1 level cut off level of <1150 pg/mL that was suitable for the identification of responders to TACE. Importantly, the identification of responders before treatment was possible for the overall population as well as in the subgroup of patients with low or normal AFP levels. This cutoff was equally suitable to identify patients with a longer survival after TACE.

DKK-1 is a soluble secreted protein consisting of 266 amino acids that blocks the formation of active Wnt-Frizzled-LRP5/6 receptor complexes, resulting in intercepting Wnt signal transduction, but is rarely expressed in normal human adult tissues except in embryonic and placental tissues [14,16,21]. It has been proposed that endothelial cells and platelets may secrete DKK-1 [22], which could explain the strong positive correlation between DKK-1 and the number of platelets (Appendix A). Due to its deep involvement in the signaling cascades of HCC development, its detectability in blood, and its molecular characteristics, DKK-1 is a promising HCC biomarker candidate. Indeed, elevated DKK-1 levels were associated with poor clinical outcome and shorter survival in previous studies [11,12,14,15]. Recent studies have shown that in healthy subjects, DKK-1 can be detected at mean serum levels ranging from 0.90 ng/mL [range, 0.01–1.91 ng/mL] to 5.9 ng/mL [23]. In Egyptian patients with hepatitis C virus infection and in Korean patients mostly positive for hepatitis B surface antigen, mean DKK-1 levels showed a two-fold elevation in patients with HCC [11,23]. Among the European HCC patients in our study, HCC was mainly based on alcohol-related liver cirrhosis (60%). The presence of liver cirrhosis was a factor influencing DKK-1 levels in serum in our cohort (Figure 2A). Thus, DKK-1 levels were two-fold lower in patients with cirrhosis (*p* = 0.002). Similarly, DKK-1 levels were described to be lower in patients with hepatitis C and cirrhosis as compared to healthy controls in a study from Egypt [23,24]. We found in our population a linear correlation with the total tumor diameter as measured by tomography (Figure 2B). Moreover, we found a significant elevation of DKK-1 levels in patients with tumor lesions ≥ 5 cm compared to patients with lesions <5 cm, similar to the findings of El-Shayeb et al. [25]. There was a linear increase in DKK-1 levels with increasing BCLC stages, reflecting tumor progression (Figure 2C). 

Treatment with TACE is a standard repertoire for treatment of HCC worldwide. However, response to TACE is heterogenous, and with the current dynamic development of more effective systemic treatments, early selection of the most effective therapy for maximizing the individual treatment success is of paramount importance [17]. Currently, AFP is widely used in clinical practice to evaluate response to HCC treatment [6]. However, AFP has not been validated for the prediction of response to TACE in HCC patients [9,10], and it cannot be used in HCC patients with normal AFP levels [14]. In our study, we could show that APF levels were associated with response to TACE (Figure 3B). However, 42/97 patients (43%) in our population had AFP levels within normal ranges (<8 ng/mL), which diminishes their value for response prediction. AFP levels showed a narrower distribution across individual patients as compared to DKK-1, which was measurable in all patients (Figure 1). In contrast to DKK-1, AFP levels showed no association with the presence of liver cirrhosis or BCLC stage. The correlation between AFP and DKK-1 was weak (r = 0.022).

DKK-1 levels were significantly higher in patients with a 12-week response to TACE as compared to refractory patients (*p* < 0.001) (Figure 3A). Hereby, DKK-1 levels showed the strongest differences between responders and refractory patients in BCLC stages A and B (*p* = 0.01 and *p* = 0.0023, respectively), but they were not different in patients in BCLC stage C/D (Figure 3D). The inability of DKK-1 levels in discriminating responders in the BCLC C/D subgroup might be caused by the small number of patients as well as by the influence of decreased liver function in those patients as compared to BCLC A/B stages. DKK-1 levels could identify 12-week responders to TACE (AUC = 0.798, *p* < 0.001) better than AFP levels (AUC = 0.679, *p* = 0.018) (Figure 3C). Importantly, DKK-1 levels were also significantly higher in patients with normal AFP levels and 12-week refractoriness to TACE as compared to responders. We could demonstrate this association of DKK-1 similarly in the subgroups of patients with AFP levels <130 ng/mL (*p* = 0.005), <20 ng/mL (*p* = 0.003), or <8 ng/mL (*p* = 0.023) (Figure 4). We find that this observation merits particular attention, as for patients with normal AFP levels, there are currently no alternative serum response markers available. We calculated an optimal cutoff for DKK-1 of 1150 ng/mL for identifying 12-week responders to TACE in the overall population of our study with a sensitivity of 78% and a specificity of 77%. The optimal DKK-1 cutoff for identification of responders was the same for the subgroups with AFP levels < 130 ng/mL (sensitivity 75%, specificity 75%) or <20 ng/mL (sensitivity 86%, specificity 77%), and it was 933 pg/mL for patients with AFP levels < 8 ng/mL with a sensitivity of 100% and a specificity of 60%. With respect to the strong performance of DKK-1 as a response marker across patients with different AFP levels in our study, we feel that DKK-1 needs to be further investigated for its potential as a tool for navigating treatment decisions, ideally not only for TACE but also for other HCC treatments.

Our observation of an association of serum DKK-1 levels with response to TACE in European patients is in line with previously described higher DKK-1 levels in Chinese HCC patients responding to TACE reported by Wu et al. [17]. The similarity of both observations is particularly important given the different genetic backgrounds and the different types of liver disease underlying HCC of the two populations. Thus, in contrast to the Asian cohort predominantly suffering from viral hepatitis, only a small proportion of our patients had viral hepatitis, but the majority had alcohol-related liver cirrhosis (Table 1). Additionally, most of the patients reported by Wu et al. were in BCLC stage C, whereas in our study, most patients were in stages BCLC A (56%) or B (34%), thus having a smaller tumor burden (Table 1). Of note, there are crucial methodological differences between both studies that need to be taken into account. Indeed, DKK-1 could be quantified in all patients included in this study, whereas in the study by Wu et al., DKK-1 was undetectable in about 20% of patients. This different dynamic range of the assays is likely determined by the approximately 2 log pg/mL lower detection limit for DKK-1 in the assay used in our study [17]. 

In other analyses in Asian patient populations, DKK-1 was reported to be useful for HCC diagnostics [12,14]. Indeed, DKK-1 has previously been shown to outperform the sensitivity of AFP for detecting early HCCs that often show AFP levels within normal ranges [26]. Other studies showed that patients with HCC have higher circulating DKK-1 levels compared to healthy subjects, and these elevated DKK-1 expression levels are associated with a worse clinical outcome [12,14,15,17]. Accordingly, in our patient population, lower DKK-1 levels were associated with longer survival (Figure 5). Although individual survival time was likely influenced by treatments that were conducted after TACE, and causes of death were heterogeneous, the association of DKK-1 as a prognostic marker for survival of HCC patients should be further investigated.

Recently, other prognostic marker for TACE therapy outcome were described. Granito et al. proposed hypertransaminasemia after TACE (AST increase ≥46%, ALT increase ≥52% compared with baseline values) as a reliable predictor of response [27]. Moreover, T cell immunoglobulin and mucin 3 (Tim-3) was also postulated as a prognostic indicator for HCC patients undergoing TACE [28]. Patients with low Tim-3 levels after TACE correlated with poor prognosis. In contrast to our study, transaminases and Tim-3 were measured after TACE and compared to baseline values. Circulating markers that are as easy to access as DKK-1 are most suitable for improving treatment personalization in clinical practice, and DKK-1 and those other markers should be evaluated for their use for response prediction in prospective studies. Our study has some limitations. First, the international community lacks a consensus definition of TACE refractoriness. The influence of possible disruptive factors for DKK-1 quantification in HCC patients needs to be clarified in future studies. Owing to the real-world character of our study, disease stages of the patients vary across different BCLC stages. Thus, prospective studies with more homogenous populations should be conducted to validate our findings.

## 5. Conclusions

In conclusion, serum levels of DKK-1 are a sensitive biomarker for tumor size and for the efficacy of TACE in European patients with early or intermediate stage HCCs. Importantly, DKK-1 levels are suitable to identify responders to TACE in HCC patients with low AFP levels, making DKK-1 a potential candidate to close the diagnostic blind spot that exists in those patients. Future studies will be necessary to validate the potential of DKK-1 as a clinical marker for HCC patients in different therapeutic settings. Our work could help to advance personalization in HCC patient care and can give impulses to investigate the mechanisms underlying treatment response in HCCs.

## Figures and Tables

**Figure 1 cancers-14-04807-f001:**
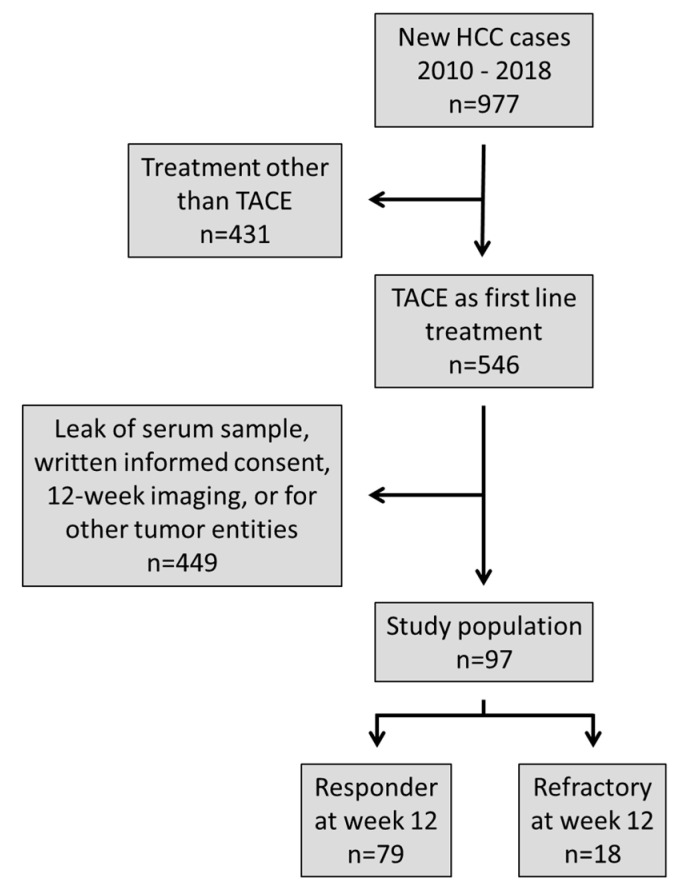
Selection of the study population.

**Figure 2 cancers-14-04807-f002:**
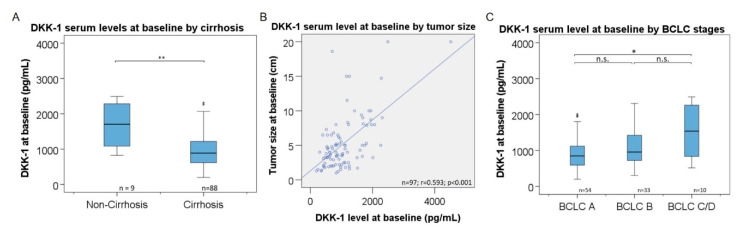
Association of DKK-1 levels with disease stage. Serum levels of DKK-1 before TACE by presence of liver cirrhosis (**A**), total radiologic tumor size (**B**), and BCLC stages (**C**). The upper and lower ends of the bar indicate the 75- and 25-percentile, respectively. The marking in the middle of the bar shows the median. Not all extreme outliers are shown. The line in (**B**) represents linear regression. * = *p* < 0.05; ** = *p* < 0.01 (Mann–Whitney test); n.s. = not significant; r = Pearson correlation coefficient of r; n = number of patients; BCLC= Barcelona Clinic Liver Cancer.

**Figure 3 cancers-14-04807-f003:**
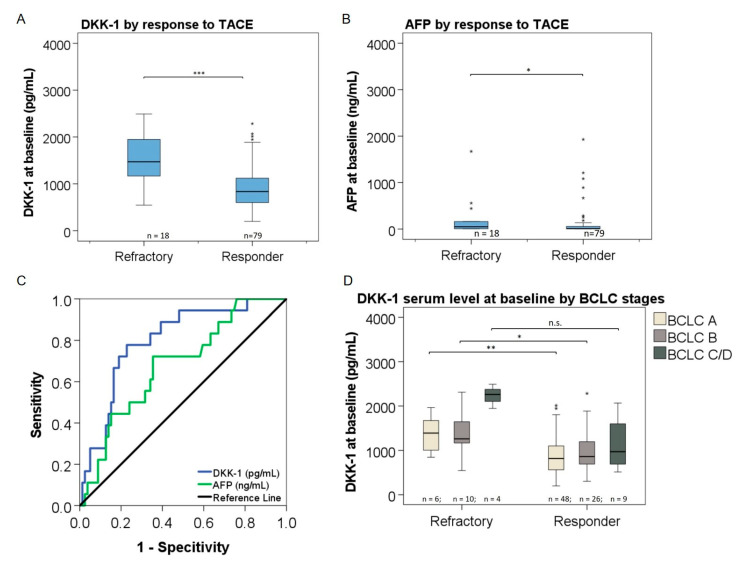
Association of DKK-1 and AFP with response to TACE. Serum levels of DKK-1 (**A**) and AFP (**B**) at baseline by 12-week response to TACE and receiver operating characteristic (ROC) analysis of sensitivity and specificity of DKK-1 and AFP for identifying responders (**C**). Serum levels of DKK-1 in patients with different BCLC stages by response to TACE (**D**). The upper and lower ends of the bar indicate the 75- and 25-percentile, respectively. Extreme outliers are not displayed. The marking in the middle of the bar shows the median. * = *p* < 0.05; ** = *p* < 0.01; *** = *p* < 0.001 (Mann–Whitney test); n.s. = not significant; n = number of patients; BCLC= Barcelona Clinic Liver Cancer.

**Figure 4 cancers-14-04807-f004:**
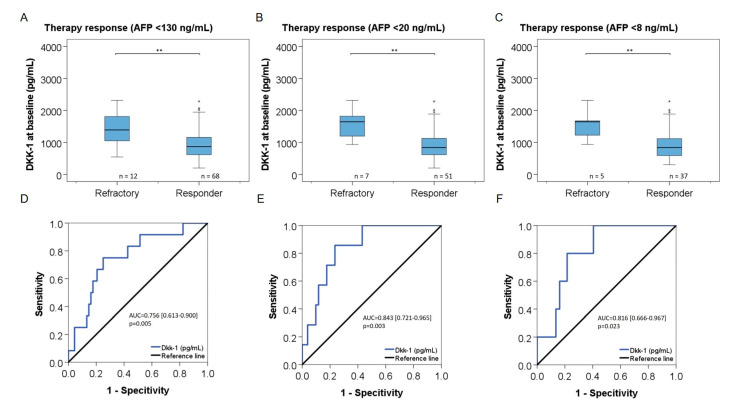
Association of DKK-1 with response to TACE in patients with low AFP levels. Serum levels of DKK-1 by 12-week response to TACE and receiver operating characteristic (ROC) analysis of sensitivity and specificity of DKK-1 for identifying responders in patients with AFP <130 ng/mL (**A**,**D**), AFP <20 ng/mL (**B**,**E**), or AFP <8 ng/mL (**C**,**F**), respectively. The upper and lower ends of the bar indicate the 75- and 25-percentile, respectively. Extreme outliers are not displayed. The marking in the middle of the bar shows the median. ** = *p* < 0.01 (Mann–Whitney test); n = number of patients.

**Figure 5 cancers-14-04807-f005:**
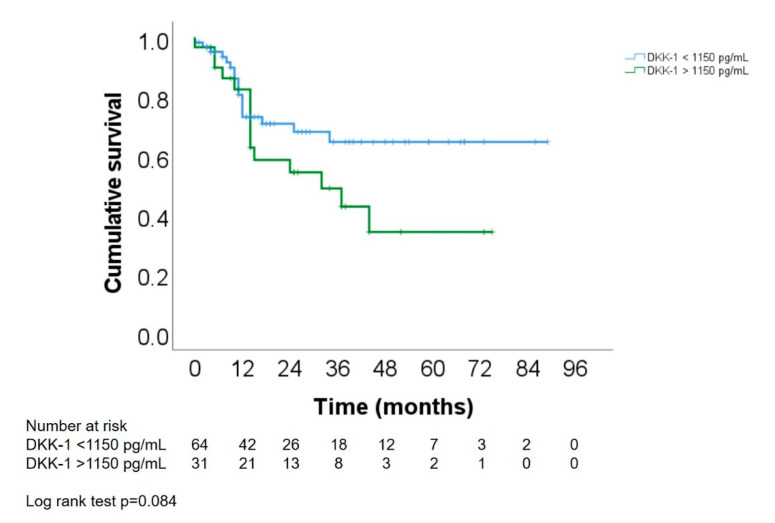
Survival analysis of the serum DKK1 levels with overall survival of patients with HCC after TACE. The patients were divided into a high and a low DKK-1 expression group using a cutoff value of 1150 ng/mL. Using log rank test, the two groups were statistically not significantly different (*p* = 0.084).

**Table 1 cancers-14-04807-t001:** Characteristics of the study population before TACE.

Characteristic	Study Population (n = 97)
	n (%)
Sex (male)	86 (89)
Age (years) *	63 ± 8.8 [range, 31–83]
Liver cirrhosis	88 (91)
**Underlying liver disease**	
Viral hepatitis	15 (17)
Alcohol-related cirrhosis	53 (60)
NASH	9 (10)
others	11 (13)
**Child-Pugh-Turgott classification**	
A	68 (77)
B	17 (19)
C	3 (3)
**BCLC stages**	
A	54 (56)
B	33 (34)
C	8 (8)
D	2 (2)
**ECOG performance status**	
0	57 (59)
1	20 (21)
2	8 (8)
n.a.	12 (12)
**Primary tumor site**	
Left lobe	21 (22)
Right lobe	52 (54)
Bilobar	24 (25)
**Number of tumor lesions**	
1	50 (52)
2	26 (27)
3	12 (12)
>3	9 (9)
**Total tumor diameter (mRECIST)**	
<3 cm	29 (30)
3–5 cm	39 (40)
6–10 cm	22 (23)
>10 cm	7 (7)
Extrahepatic HCC at baseline	31 (32)
DKK-1 (pg/mL) *	1073 ± 636 [range, 199–4514]
AFP (ng/mL) *	388 ± 1628 [range, 1.4–13019]
Albumin (g/L) *	38 ± 6 [range, 18.6–49]
ALT (µkat/L) *	0.8 ± 0.7 [range, 0.16–5.7]
Bilirubin (µmol/L) *	25 ± 27 [range, 4–189]
CRP (mg/L) *	8.9 ± 12.5 [range, 0.3–78]
Leukocytes (exp^9^/L) *	6.2 ± 2.5 [range, 1.4–15.6]
Neutrophils (exp^9^/L) *	4.0 ± 2.1 [range, 0.62–12.9]
Platelets (exp^9^/L) *	149 ± 97 [range, 40–531]

* = mean ± standard deviation [range]; BCLC = Barcelona Clinic Liver Cancer; ECOG = Eastern Cooperative Oncology Group; mRECIST = modified Response Evaluation Criteria in Solid Tumors; n.a. = not applicable; ALT = Alanine Aminotransferase; CRP = C-reactive protein.

**Table 2 cancers-14-04807-t002:** DKK-1 levels in patients with low AFP and diagnostic accuracy of DKK-1 in classification of 12-week response to TACE.

AFP Subgroups	DKK-1 *		*p*-Value	AUC [95% CI]	DKK-1 Cutoff	Sensitivity	Specificity
	Refractory	Responder					
AFP < 130 ng/mL (n = 80/97)	1391 [546, 2313] pg/mL (n = 12)	868 [199, 2284] pg/mL (n = 68)	0.005	0.756 [0.613, 0.900]	1150 pg/mL	75%	75%
AFP < 20 ng/mL (n = 58/97)	1646 [936, 2313] pg/mL (n = 7)	840 [199, 2284] pg/mL (n = 51)	**0.003**	0.843 [0.721, 0.965]	1150 pg/mL	86%	77%
AFP < 8 ng/mL (n = 42/97)	1646 [936, 2313] pg/mL (n = 5)	840 [305, 2284] pg/mL (n = 37)	**0.023**	0.816 [0.666, 0.967]	933 pg/mL	100%	60%

* = median [range]; n = number of patients.

## Data Availability

The data presented in this study are available in this article and supplementary material. Individual participant data pertinent to the results reported in this publication will be shared after de-identification to researchers who provide a methodologically sound and ethically approved research proposal. To gain access, data requestors will need to sign a data-access agreement.

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
