# Peer review of "Dickkopf-Related Protein 1 as Response Marker for Transarterial Chemoembolization of Hepatocellular Carcinomas"

_cancers, 2022, doi:10.3390/cancers14194807_

Round 1

Reviewer 1 Report

The paper is interesting and well written. It is focuse on a new prognostic markers in HCC patients undergoing TACE.

I would recommend to comment also other prognostic facters recently identified, for example transient hypertransaminasemia (cite the paperPMID: 34683182 )

Please add the number at risk in the survival curves

Reviewer 2 Report

In this manuscript, the authors evaluated the utility of DKK1 as a response marker for TACE in European HCC patients. This is the retrospective single-center small cohort analysis and there seems to be no marked utility of DKK1 as a biomarker in overall survival. At this time, this manuscript is not able to be accepted for publication because some concerns remain.

1.     In Fig 5, there does not appear to be a significant difference in overall survival up to week 20 (unlike the Asian patient report). Is there a significant difference in this survival curve? What is the reason for the different prognosis after 20 weeks after TACE and is DKK-1 a useful predictive marker for TACE treatment?

2.     Although difference in DDK1 level has been observed between responder and refractory, if this is simply a reflection of BCLC stage, such as tumor size and number of tumors, it would be of little significance to measure DKK1. It is desirable to analyze whether DKK1 reflects malignant nature of HCC other than tumor size and number in a multivariate analysis including confounding factors in order to consider the clinical significance of DKK1.

Reviewer 3 Report

The authors used Dickkopf-related protein 1 (DKK-1) as a novel marker for predicting transarterial chemoembolization (TACE) response in hepatocellular carcinoma (HCC) patients. This DKK-1 showed superior result than conventional alpha fetoprotein (AFP). Since there are no clear predictive markers of TACE-refractoriness, this study suggests that DKK-1 is one of the potential candidates for predictors of TACE-refractoriness, particularly in patients with low AFP. However, there are critical concerns in this study.

Major points:

1. From the study design, it was necessary to divide the study population into derivation and validation cohorts to confirm reproducibility. However, only the derivation cohort was established in this study. Establishing independent validation cohort is necessary.

2. As most patients were excluded from this study, correlation between DKK-1 and baseline characteristics needed to be interpreted with caution, as potential selection bias may arise, but some statements have been rushed to conclusions. For example, most non-cirrhotic HCC patients would have been treated with surgical resection and only patients with large or aggressive tumors had been selected in this study, which has high DKK-1. This may have resulted in higher DKK-1 levels in non-cirrhotic patients than in cirrhotic HCC patients in this study.

3. Some patients who are not suitable for TACE (e.g., BCLC stage D or Child–Pugh class C) are included in this study. Within these patients, due to the risk of hepatic failure, HCC treatment with TACE could be limited. It seems inappropriate to include these patients in this study.

4. The statistical evaluation of subpopulation with low AFP seems odd. As a subgroup analysis, it seems quite odd to subgroup patients with overlapping values (AFP <130 ng/mL, AFP <20 ng/mL, and AFP <8ng/mL). Moreover, the rationale for deciding cutoff value of AFP under 130 ng/mL seems arbitrary. Further evidence is needed to support this cutoff point.

Minor Points:

1. Please correct ‘alfa fetoprotein’ in the simple summary paragraph to ‘alpha fetoprotein’.

2. There is no P value in neither Figure 5 nor the manuscript. Please provide the result.

Round 2

Reviewer 2 Report

Authors have correctly addressed the concerns.

Reviewer 3 Report

I believe the authors have responded appropriatedly to Reviewers' comments.